# The Uptake of Engineered Nanoparticles by Sludge Particulates

**Soohoon Choi** [1,2,*], **Murray Johnston** [3], **Gen-Shuh Wang** [4] and **Chin-Pao Huang** [1]

1. Department of Civil and Environmental Engineering, University of Delaware, Newark, DE 19711, USA
2. Department of Environmental Engineering, Chungnam University, 99 Daehak-ro, Yuseong-gu, Daejeon 34134, Republic of Korea
3. Department of Chemistry and Biochemistry, University of Delaware, Newark, DE 19711, USA
4. Institute of Environmental Health, National Taiwan University, Taipei 100, Taiwan; gswang@ntu.edu.tw
* Correspondence: crimson@cnu.ac.kr

**Abstract:** The aim of the study was to understand the removal characteristics of engineered nanoparticles (ENP) from sludge treatment processes in wastewater treatment plants (WWTP). Removal of ENP ($TiO_2$, $ZnO$) was tested on primary and secondary sludge, using differential sedimentation experiments to quantify the attachment of ENP to sludge particulates. To better understand the attachment characteristics, aquatic conditions such as mixed liquid suspended solid concentration, and Ionic strength of the wastewater, were varied to replicate different field conditions of WWTPs. Results showed different degrees of multilayer attachment to sludge surfaces based on the experimental conditions. To verify the effect of ENP surface characters with the sludge attachment, $SiO_2$, $ZnO$, and $TiO_2$ were tested, showing $SiO_2$ with the highest amount of attachment regardless of its surface charge. With the variation of sludge concentration, up to four degrees of magnitude in sorption was observed. Salt concentrations also showed high impacts on the sorption, where the sorption is decreased by half when doubling the salt concentration. The findings of the current research may aid in understanding the fate of engineered nanoparticles in wastewater treatment plants.

**Keywords:** ionic strength effect; mixed liquid suspended solid concentration; nanoparticle; sludge particulates; sorption

## 1. Introduction

With the increased use of engineered nanoparticles (ENPs) in consumer products, the release of ENPs into municipal wastewater systems is inevitable. Nanoparticles in the municipal wastewater stream have been proven to interact with organic fragments in the sewer system, aiding nanoparticle transport to wastewater treatment plants (WWTP) without substantial particle loss [1]. The ENP that flows into the WWTPs has also been reported as impacting the performance and characteristics of wastewater sludge through its interactions with organic and microbial matter. $TiO_2$ and $ZnO$ have been reported to induce negative impacts on wastewater sludge, where it has been reported to reduce the flocculation ability, resulting in the alteration of the size and structure of the flocs, and also reducing the dewaterability of sludge [2–4]. $SiO_2$ has additionally been reported to have a negative effect on the settleability of sludge, and an increase in fouling in membrane bioreactors [5,6]. This is due to the fact that contact of ENPs with microbial substances induces extracellular polymeric secretion, their mainly being high molecular weight substances, such as protein and amino-like substances [7–9]. These high molecular weight substances reduce the flocculation kinetics of the sludge, resulting in smaller and less stable flocs. Studies have shown a decrease in the size of sludge flocs with $TiO_2$ inflow, and a higher concentration of effluent-suspended solids with $ZnO$, indicating a negative impact on WWTP performances with the inflow of ENPs [2,3]. In addition to the impacts on sedimentation, ENP has also been known to negatively impact microbial properties. It has been proven that with the contact of $TiO_2$, $ZnO$, and fullerene, the number of reactive

oxygen species and lactate dehydrogenase was increased, indicating higher stress levels in the sludge [10–12], where the exposure to $SiO_2$ has shown a depression in nitrogen removal efficiency [13]. The microbial diversity within the sludge has also been known to decrease with ENP contact, where the decrease has been reported to occur within 360 min of exposure [10]. The combined impact of ENP has been reported to reduce nutrient removal, including biochemical oxygen demand (BOD), nitrogen, and phosphorous [12,14,15].

An additional influence of WWTP on the fate and transport of ENPs is its redistribution to secondary locations. It has been estimated that 26~39% of ENPs discharged into sewage systems flow to wastewater treatment plants, which are later distributed to secondary treatment facilities or ultimately discharged to the environment [16]. The majority of ENPs discharged to the WWTP were detected in the primary and the secondary sludge mass [17–19] due to the accumulation of ENPs on the sludge particulates. In the United States, it is estimated that over 50% of WWTP sludge is processed into biosolids for fertilizer [20]. With the biosolids, it has been reported that high concentrations of metallic species have been detected, where titanium has been detected in concentrations of up to 7020 mg $TiO_2$/kg dry solids, 100 mg $SiO_2$/kg dry solids, and zinc at concentrations up to 8550 mg Zn/kg dry solids [21,22]. The metals may impact the soil if the land application of biosolids is practiced, where reports show that nano-$TiO_2$, $SiO_2$, and nano-ZnO can reduce microbial biomass and diversity or alter the soil metabolite profile [23,24]. It is also known that the presence of ENPs in the soil environment can influence the biogeochemical nitrogen cycles and increase carbon emissions [25,26]. Nano-ZnO, having relatively higher solubility with respect to nano-$TiO_2$, has imposed a much stronger stress on the rhizosphere than nano-$TiO_2$ [27,28]. To better understand the fate of ENPs and their impacts on the environment, the interactions between ENPs and sludge particulates in WWTP should be assessed further.

The main goal of this research was to study the interactions between ENPs and Sludge under various conditions. The extent of ENPs attachment to sludge particulate and its constituents were studied by determining the number of ENPs sorbed as a function of free ENPs concentration (i.e., sorption isotherms) using a "differential sedimentation" method. Sampled sludge from the primary and secondary sedimentation tanks of a wastewater treatment plant was used to conduct the experiments to understand the fate of ENP throughout the wastewater treatment process. To further understand the attachment of ENPs to sludge, experiments with $TiO_2$ and ZnO, due to their high detection in wastewater sludge, were conducted under various aquatic conditions. Initially, the sludge was divided into four separate fractions, dissolved organic matter (DOM), small organic particular matter (SOPM), large organic particular matter (LOPM), and whole sludge mass (WSM). The sludge concentration and conductivity were also varied to understand the degree of ENP attachment under different field conditions. Additionally, ENPs of different compositions were also tested to understand the effects of surface chemistry on the attachment characteristics.

## 2. Materials and Methods

### 2.1. Preparation of Engineered Nanoparticles and Sludge Particulates

Three types of nanoparticles, i.e., $SiO_2$, $TiO_2$, and ZnO, were selected in this study because of their large consumer market. Commercial grade, nano-sized $TiO_2$ (>99.5%) was purchased from Degussa Corp. (Degussa Aeroxide P25, Parsippany, NJ, USA) and was used as received without pre-treatment. AEROXIDE P25 belongs to the mixed crystal type, including anatase, rutile, and a small amount of amorphous $TiO_2$ [29,30]. ZnO nanopowder (<50 nm) was purchased from Sigma Aldrich, Saint Louis, MO, USA, (>97%) and used as received. Silica dioxide was purchased from Sigma Aldrich, in the form of fumed amorphous silica (Aerosil 200, $d$ = 0.2–0.3 μm). Prior to each experiment, nanoparticle suspensions were prepared fresh with deionized water and sonicated with a high prior to each experiment; nanoparticle suspensions were prepared fresh with deionized water and sonicated with a high-powered sonicator at 100 W for 10 s (Ultrasonic Homogenizer

4710 Series, Cole-Parmer Instrument Co., Chicago, IL, USA) before introducing to the batch reactors.

For sorption experiments, sludge samples from the primary and secondary sedimentation basins of the Wilmington Wastewater Treatment Plant in the City of Wilmington, DE, USA were used. The sludge samples were firmly capped and stored in a temperature (dried ice-packed)-controlled chamber while being transported to the laboratory. The sludge samples were stored in a refrigerated environment of 4 °C before experiments. Experiments were conducted within three days after sampling.

The mixed liquid–suspended solid (MLSS) concentrations, as well as the elemental composition of the sludge, were tested based on the 2540D method in Standard Methods for the Examination of Water and Wastewater, before the experiments. Results showed an MLSS concentration of 2000~4000 mg/L for the secondary sludge and 700~1500 mg/L for the primary sludge. The organic content of the primary sludge was tested to be at an average of 90% with the organic content of the secondary sludge to be at an average of 77%. Elemental analysis using an ICP-OES (Agilent Technologies Inc. 725, Santa Clara, CA, USA) showed a negligible amount of titanium and zinc in the sludge (<0.85 μg/L for both Ti and Zn).

To understand the interactions of ENPs and sludge particulates, sludge particulates were separated into four different size fractions: (1) dissolved organic matters (DOMs), size < 100 nm, (2) small organic particulate matters, SOPMs, (100 nm < $d$ < 1 μm), (3) large organic particulates, LOPMs, ($d$ > 1 μm), and (4) whole sludge, mass (WSM) containing all components indicated above; that is, these are the sludge samples that receive no separation process.

The DOM fraction was obtained by centrifugation (20 mL) at 20,000 g for 40 min in a high-speed centrifuge (Marathon 22 k, Fisher Scientific, Waltham, MA, USA), repeated three times for maximum separation and the center collected. The second fraction, SOPM, containing mostly DOMs and sludge particles smaller than 1 μm in diameter, was obtained through gravity sedimentation in an Imhoff cone for 4 h. The supernatant was then collected as SOPM, which includes the DOMs and sludge particles smaller than 1 μm. The third fraction, LOPM, consisting of organic particle matters larger than 1μm in diameter and supposedly free of any DOMs or small organic particulates, was collected gravimetrically as above. After sedimentation for 4 h, the supernatant was removed, and the Imhoff cone was filled with DI water and let settle again for 4 h to remove the SOPMs. The above steps were repeated at least four times to remove all DOMs and particles smaller than 1 μm from the sludge. The fourth fraction was the total original sludge mass, after filtration through a 4.5 mm ASTM standard sieve to remove any debris [31].

To understand the effects of ionic strength on ENP to sludge sorption, various ionic strength conditions were also tested. Sampled sludge was bagged in seamless cellulose dialysis tubes with a molecular cutoff of 1 kDa. The dialysis tube was submerged in a two-liter tank receiving a continuous flow of deionized water (18 mΩ) at 0.1 mL/min for 12 h under constant stirring with a magnetic stirrer. Deionized water was continuously circulated into and out of the tank until the sludge reached conductivity values under 0.26 mΩ. Ionic strength was controlled with a NaCl stock solution (2000 mM), where the pH was controlled with HCl and NaOH (Sigma-Aldrich, analytical grade).

### 2.2. Differential Sedimentation

To measure the attachment of ENPs to various materials, there are a limited number of methods developed and currently in use. For example, batch sorption experiments are a widely used method, where nanoparticles and sorbents such as sand and soil [32,33] are agitated in overhead shakers, and later separated through the centrifuge. Column experiments are also a popular method where the nanoparticles are injected continuously or in a pulse method, and sorption parameters are measured from the difference of the inflow and outflow concentration [34–36]. Currently, quartz microbalance has been used

as a novel technique to verify the sorption based on the frequency shift of particles to the quartz surface [37,38].

However, the experiment mentioned above is based on the attachment of a mobile substance (ENP) onto stationary material. To tackle this issue, the current research focuses on a differential sedimentation process, where the free particle concentration is extrapolated based on the sediment characters of suspended particles. The process is conducted by mixing sludge and nanoparticles together and left to settle, while the turbidity of the sample is continuously measured. After the flocs settle, the turbidity data from the suspended ENP show a linear decrease in their concentration. The turbidity of the sedimentation process was measured with a UV-visible spectrophotometer (DR5000, HACH, Loveland, CO, USA) to measure the light transmittance and converted to turbidity. Based on the time-dependent concentration decrease of the free particles, a sedimentation curve can be extrapolated, where the free particle concentration before floc sedimentation can be calculated. (Figure 1). Sedimentation curves of the sludge–ENP mixture were analyzed following the approach developed by Nicolosi et al. (2005) who describe the local concentration as a function of time [39].

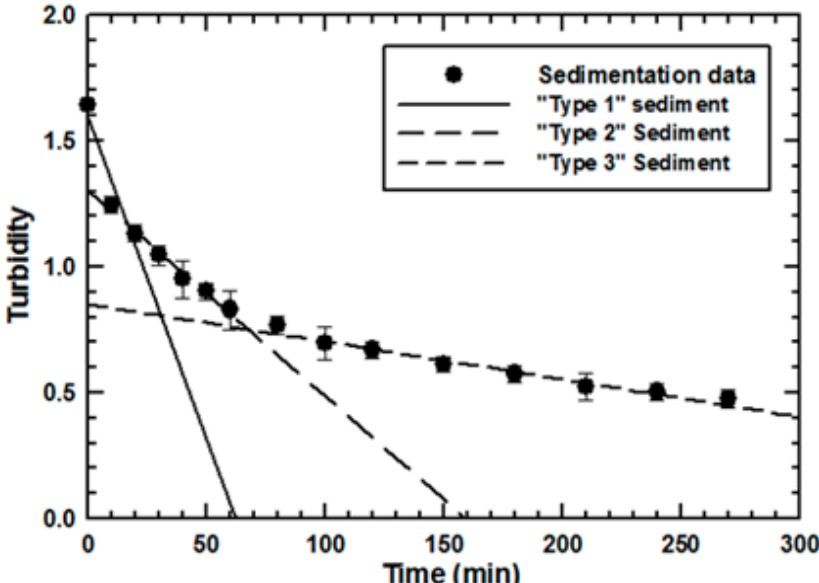

**Figure 1.** Sedimentation curve and analysis of ENP to sludge. Conditions: pH 6.5, ZnO concentration = 80 mg/L, Sludge = Wilmington Primary sludge.

Based on the local conservation equation for mass and linear momentum, the local concentration of the settling particles can be expressed as

$$C_t = C_o e^{-t/\tau} \tag{1}$$

where $C_t$ is the local concentration at the sedimentation time $t$; $C_o$ is the concentration at the beginning of the sedimentation; $\tau$ is the time constant, which is a hydrodynamic-chemical characteristic property of particles. Based on the spherical shape of the sediments, the time constant can be calculated by the following equation.

$$\tau = \frac{9\eta\left(\beta - p_f\right)}{2g^2(\rho_s - \rho_l)^2 R_H^2} \tag{2}$$

where $\eta$ is the viscosity, $\beta$ is related to the solid–fluid interaction force and has the dimension of pressure, $p_f$ is the fluid pressure, $g$ is the acceleration due to gravity, $\rho_s$ is the solid component density, $\rho_l$ is the liquid density, and $R_H$ is the hydrodynamic radius. Due to the size and characteristics of the organic flocs, the sedimentation process can be separated into

three phases. Each phase occurs based on the size of the flocs, where large and small flocs show different sedimentation behavior. After ENP sorption to sludge reaches equilibrium, three different sediment types can be observed, with large agglomerates of organic flocs and ENP, followed by small sludge particulates observed with ENP. Additionally, nanoparticles that do not sorb to sludge surfaces also exist in the mixture as well. This can be characterized in the sedimentation curve, where three asymptotic lines can be drawn, each representing a different sediment type. In the current research, each sediment type will be characterized as "type 1" and "type 2" aggregates for the large and small sediments, respectively. Free ENP particles are named "type 3" particles, for the sake of differentiation.

Although the turbidity of each sediment type cannot be measured, the total turbidity of the sedimentation process has been measured as a function of time and represented as $T_{Total}(t)$. The total turbidity is a summation of each type of sedimentation ("type 1", "type 2", and "type 3") where each type is indicated as $T_1(t)$, $T_2(t)$, and $T_3(t)$.

$$T_{total}(t) = T_1(t) + T_2(t) + T_3(t) \tag{3}$$

Turbidity can be related to concentration by its extinction coefficient $\alpha$, which is assumed constant throughout the sedimentation experiment.

$$T(t) = \alpha C(t) \tag{4}$$

By combining Equations (3) and (4), the sedimentation equation can be rewritten as

$$T_{total}(t) = \alpha_1 C_1(t) + \alpha_2 C_2(t) + \alpha_3 C_3(t) \tag{5}$$

$$T_{total}(t) = \alpha_1 C_1(0)e^{-\frac{t}{\tau_1}} + \alpha_2 C_2(0)e^{-\frac{t}{\tau_2}} + \alpha_3 C_3(0)e^{-\frac{t}{\tau_3}} \tag{6}$$

$$T_{total}(t) = T_1(0)e^{-\frac{t}{\tau_1}} + T_2(0)e^{-\frac{t}{\tau_2}} + T_3(0)e^{-\frac{t}{\tau_3}} \tag{7}$$

where $\alpha_1$, $\alpha_2$, and $\alpha_3$ are the extinction coefficient of type 1, 2, and 3 sedimentations, and the zero in the quotation mark means the beginning of the sedimentation.

The term $T_{total}(t)$ can be measured using a UV-visible spectrophotometer, whereas $T_1(t)$, $T_2(t)$, and $T_3(t)$ is calculated. The difference in the settling properties of "type 1", "type 2", and "type 3" is the priority requirement for their successful separation from each other during data analysis. The difference in sedimentation will allow the determination of the time constants, i.e., $\tau_1$, $\tau_2$, and $\tau_3$ from which $T_1(0)$, $T_2(0)$, and $T_3(0)$ can be calculated. The "separation time" (time at which the slope of the sedimentation changes) from "type 1" to "type 2" aggregates are designated as $t^*_{12}$, and $t^*_{23}$ from "type 2" to "type 3". Since "type 1" sedimentation proceeds ahead of "type 2", $t^*_{12}$ always arrives before $t^*_{23}$. Intuitively, when the sedimentation time reaches $t^*_{12}$, it means that "type 1" aggregates have already settled and $T_1(t)$ becomes zero (Figure 1). When the sedimentation time reaches $t^*_{23}$, "type 2" aggregates completely settle; there, both $T_1(t)$ and $T_2(t)$ are zero. (Figure 1).

$$T_{total}(t') = T_3(0)e^{-\frac{t}{\tau_3}} \tag{8}$$

$$ln\,T_{total}(t') = \ln T_3(0) = t'/\tau_3 \tag{9}$$

Now, we can go back to calculate $T_2(0)$ and $\tau_2$. When the sedimentation time reaches $t^*_{12}$, "type 1" aggregates already settle and $T_1(t)$ is zero. Therefore, when $t = t'$ $(t > t^*_{12})$, $T_2(0)$ and $\tau_2$ can be obtained according to Equation (12).

$$T_{total}(t') = T_2(t') + T_3(t') \tag{10}$$

$$T_{total}(t') - T_3(t') = T_2(0)e^{-t/\tau} \tag{11}$$

$$\ln\left(T_{total}(t') - T_3(t')\right) = \ln T_2(0) - t'/\tau_2 \tag{12}$$

Finally, $T_1(0)$ and $\tau_1$ can be obtained using Equation (13). Figure 1 shows an example that the turbidity data are well fitted by using this procedure.

$$\ln(T_{total}(t) - T_3(t) - T_2(t)) = \ln T_1(0) - t\prime/\tau_1 \tag{13}$$

### 2.3. Determination of Free-Engineered Nanoparticle Concentration

By using the calibration curve of turbidity versus ENP concentration, $T_3(0)$ is used to calculate free ENP concentration, which represents ENP equilibrium concentration, $C$ (mg/L). ENP uptake $\Gamma$ (#TiO$_2$ per kg sludge) was calculated with the mass balance equation

$$\Gamma = \frac{c_o - c}{X} \tag{14}$$

where $c_o$ is the initial concentration of ENP (mg/L), and $X$ is the organic concentration of sludge (g/L). The free ENP concentration was extrapolated by the turbidity value of $T_3(0)$, using the intercept of the y-axis and the trend line of $T_3(0)$. Once the $T_3(0)$ value was obtained, the free ENP concentration was extrapolated from a calibration curve representing the relationship between the turbidity and ENP concentration. The calibration curve of ENP concentration was obtained under various DOM concentrations. For accurate conversions, the total organic carbon (TOC) values of the samples after the sedimentation experiments were also measured for their dissolved organic concentration (Teledyne Tekmar, Mason, OH, USA, Apollo 9000HS).

### 2.4. ENP Sorption Experiment and Evaluation

For the sedimentation test, ENPs of various amounts were introduced into the sludge samples. To obtain a sediment curve a time-dependent measurement of turbidity should be conducted. Samples were mixed in 12 mm disposable polystyrene cuvettes, where sludge and ENP were added along with Di water or NaCl, depending on the experiment. The cuvettes were shaken and placed on a shaking plate for 2 h. The cuvettes were then placed in a UV-visible spectrophotometer (HACH DR5000) to measure the light transmittance at the wavelength of 600 nm. The data was converted to turbidity, $T$, using the Lambert–Beer law.

$$\frac{I}{I_o} = e^{-Tl} \tag{15}$$

where $I/I_o$ is the transmittance, $T$ is the turbidity and $l$ is the sample length. The light transmittance was recorded every 10 min for the first 1 h, 20 min for the next 1 h, and measured every 30 min for the next 2 h. Each experiment was repeated 5 times for accuracy.

The Freundlich model was chosen to describe the uptake process as this model is not restricted to a monolayer case and it avoids the assumption of surface homogeneity. The Freundlich model has the form:

$$\Gamma = K_F C^{1/n_F} \tag{16}$$

where $\Gamma$ is TiO$_2$ uptake (ppm-TiO$_2$/g-Biomass), $C$ is TiO$_2$ equilibrium concentration (mg/mL), and $K_F$ is the Freundlich constant (mL/g-Biomass). The parameter $1/n_F$ is related to the sorption intensity and $1/n_F$ is between zero and one [40]. To determine $K_F$ and $1/n_F$, data are fitted to the logarithmic form

$$\ln\Gamma = \ln K_F + 1/n_F \ln C \tag{17}$$

Similar isotherms have been observed for other associations of nanoparticles with cells [41,42].

Due to the multiple layer sorption of ENP on sludge particulates the Brunauer–Emmett–Teller (BET) analysis was also used. The BET equation has the form.

$$\Gamma = \Gamma_m \frac{K_1 C_{eq}}{\left(1 - K_2 C_{eq}\right)\left(1 - K_2 C_{eq} + K_1 C_{eq}\right)} \tag{18}$$

where $K_1$ is the equilibrium constant of ENP adsorption for the first layer, $K_2$ is the equilibrium constant of adsorption for upper layers, $C_{eq}$ the equilibrium concentration of ENP, and $\Gamma_m$ the monolayer adsorption capacity of the adsorbent.

## 3. Results and Discussion

As has been revealed in previous studies, the bulk of the ENP is found in the primary and secondary sludge [22,23]. From this, it can be deduced that the fate of ENP in WWTP primarily relies on the sorption of ENP to DOM and organic particulates. So, based on the established method, this study will focus on the interaction of $TiO_2$ and ZnO with organic particulates. Factors that influence the attachment, such as ionic strength, concentration ratio of organic particulates to ENP, the composition of ENP, and characteristics of the organic particulates were investigated. The various parameters selected are to represent practical and realistic conditions of a WWTP. Based on this study, it will be possible to understand and predict the fate of ENP due to attachment in the primary and secondary sediment tanks.

### 3.1. Effect of Sludge Particulate Size on Engineered Nanoparticle Sorption

Sedimentation experiments were conducted with ZnO and $TiO_2$ nanoparticles adsorbed to primary and secondary sludge (Figures S1–S4). As mentioned in the methodology above, sludge was partitioned into four fractions based on the particle size of the organic material. Each fraction was separately tested with various concentrations of ZnO and $TiO_2$ (0, 10, 20, 40, 60, and 80 ppm). Note that the pH and ionic strength of all the samples were matched to that of the initial condition of the sampled WWTP.

Figure 2a shows the results of the sedimentation experiment conducted with $TiO_2$ and DOM. As it has been proven in previous studies, the interaction of DOM with ENP resulted in higher particle stability [43,44]. This is seen in Figure 2a, where the turbidity of the sludge and $TiO_2$ (TS) did not change with time, indicating little to no sedimentation during the experimental process. Similar results were also observed with the interaction between ZnO and DOM from primary and secondary sedimentation tanks, where "type 3" sedimentation (free particles) composed a majority of the results (Figure S1b,c). Due to the results, it can be speculated that the DOM (which is smaller in size compared to nanoparticles, <10 nm) formed a coated layer on the particle surface, resulting in a steric repulsion layer aiding in the nanoparticle stability.

The second series of sediment experiments, shown in Figure 2b, was conducted on small organic particles (<1 μm) and DOM. Larger organic particles (>1 μm) were settled and removed from the samples to verify the effects of suspended organic matter on the attachment and fate of ENP. Results mainly showed a change in the sedimentation characteristics, showing a steeper inclination in the initial stages (0~100 min) flowed by a moderate inclination (100~280 min). The two curves were "type 2" and "type 3" sedimentation, indicating the removal of nanoparticles through initial adsorption and sedimentation with the residual of suspended particles. Similar results were also observed with secondary sludge; however, secondary sludge exhibits a sharper initial turbidity drop, compared to the primary sludge (Figure S2a). This is due to the characteristics of the sludge, where primary sludge is mainly consisted of inorganic granular, and dissolved organic material, and secondary sludge is mainly consisted of organic flocs. With primary sludge, a portion of the constituents, which were inorganic granular, matter was removed leaving only the small organic matter portion of the constituents. Since the faster-settling inorganic granular matter was removed, the comparatively slower-settling organic matter remained in the current samples, resulting in a more gradual decrease of the sedimentation

curve. However, secondary sludge is composed of organic flocs and DOM, resulting in the initial "type 2" sedimentation of the flocs and the secondary "type 3" sedimentation of the DOM. The initial "type 2" sedimentation is assumed to be due to not only the flocs but the microbial matter in the samples as well. Among the particles tested, ZnO showed a higher degree of sedimentation compared to TiO$_2$ (Figure S2a,c). This may be due to the surface charge of the particles, where the positive charge of the zinc particles results in the charge screening of the organic flocs or dissolved organic matter. It can also be seen that ZnO in secondary sludge shows accelerated sedimentation of small flocs resulting in a "type 1" sedimentation in the early stages of the sedimentation process as well (Figure S2c).

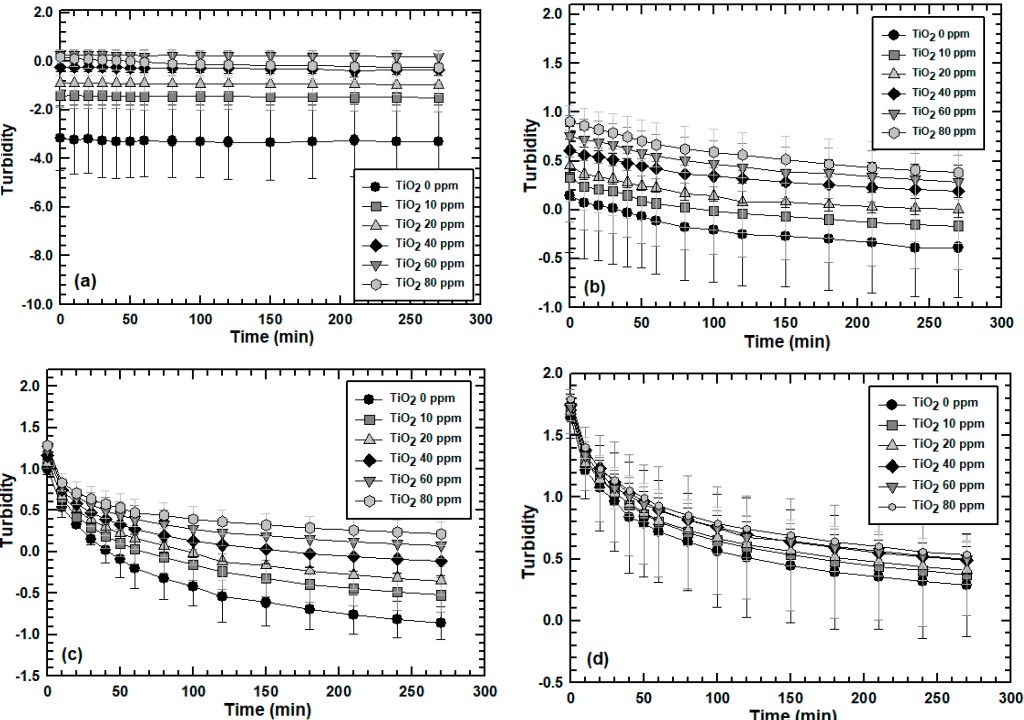

**Figure 2.** Sedimentation profile of TiO$_2$ with various organic matter fractions of primary sludge. MLSS = 2242 mg/L. Reaction time 5 h. pH = 6.8 (**a**) DOM + TiO$_2$, (**b**) Small organic matter + TiO$_2$, (**c**) Large organic matter + 2, (**d**) Sludge + TiO$_2$.

Figure 2c shows the attachment of TiO$_2$ to secondary large organic particulates matters, LOPMs (>1 μm). Due to the low concentration of SOPMs in the samples, sedimentation profiles mainly show "type 1" (0~30 min) and "type 3" (50~280 min) sedimentation. Type 2 sedimentations are visible in the 30~50 min range; however, they occur in a shorter time frame. It can also be seen that, compared with small organic matter, the residual free particle concentration is lower with the large organic particles, indicating that larger organic particles have a higher impact on ENP sorption. Additionally, regardless of the initial concentration, ZnO displays a small difference in the free particle concentrations after the sedimentation process. This indicates a strong affinity of ZnO particles due to the positive surface charge of the particles (Figure S3). With ZnO particles, a slight decrease in the "type 3" sedimentation can also be observed, indicating a small amount of coagulation of the ZnO particles. This may be due to residual DOM in the sample where it has been proven that low organic concentration may aid in the coagulation of ENP [45]

Figure 2d shows the sedimentation profile of wastewater sludge that did not undergo any separation. This profile shows sediment and adsorption characteristics of all three sedimentation types, where "type 1" sedimentation is observed in the 0~30 min area, "type 2" sedimentation is observed in the 30~210 min area, and "type 3" sedimentation is observed in the 210~280 min area. As mentioned in the previous section, the amount of turbidity decrease in the sedimentation experiments indicates the amount of ENP sorption

to the sludge particulates. With the various experiments above investigating the interactions of ENP and different sludge fractions, it may be assumed that the addition of the turbidity differences in each sludge fraction will equal the amount of turbidity drop in the total sludge. However, results show that the attachment of ENP to untreated total sludge is higher than the added amounts of the individual sludge constituents. As can be seen in Figure 2a–d, where the turbidity difference with DOM does not show any decrease, turbidity difference with small organics particulates (<1 μm) + DOM = 0.4, large organics (>1 μm) = 0.8, and un-separated field sludge (DOM + small + large sludge particulates) = 1.4. Hence, the sum of the turbidity difference of the three samples (DOM + small organic + large organic = 1.2) does not equal the total turbidity difference of the original sludge (1.4). This indicates that the adsorption of ENP to sludge is a complex process where the simple summation of sorption to DOM and sludge particulates does not represent the total sorption of ENP.

### 3.2. Effect of Engineered Nanoparticle Type: ZnO, TiO$_2$, SiO$_2$

The effect of different ENPs on sludge sorption was tested with three different nanoparticles commonly found in wastewater treatment plants. To identify the interactions of different ENPs in the inflow of wastewater treatment plants, LOPM of primary wastewater was selected as the sorbents. Figure 3 illustrates the Freundlich sorption isotherm of the three different nanoparticles (ZnO, TiO$_2$, and SiO$_2$). Results showed TiO$_2$ (●) with the lowest sorption capacity, with ZnO (▽) and SiO$_2$ (■) each displaying higher degrees of sorption of over one to two degrees of magnitude, respectively. The higher degree of ZnO sorption, compared to TiO$_2$, may be explained by the surface charge of the particles, which can be seen in the zeta potential curve in Figure 4. The zero-point charge of TiO$_2$ used in the experiment was measured to be approximately 6.5, agreeing with the corresponding crystalline composition (anatase 70% and rutile 30%) of the particles [46]. Under pH 6.2 (for which the conditions of the wastewater were samples and experiment was conducted), ZnO (○) shows a near +10 mV where TiO$_2$ (●) shows a near neutral charge.

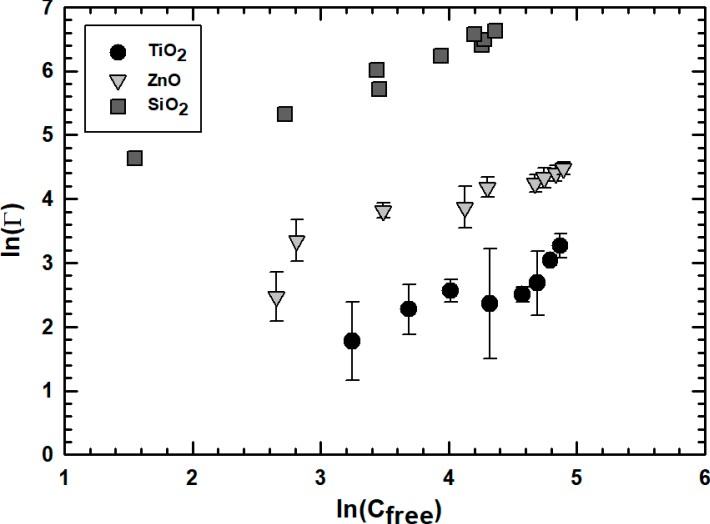

**Figure 3.** Sorption of engineered nanoparticles composed of various elements. Large organic particular matter (LOPM), MLSS = 2242 mg/L. Reaction time 5 h. pH = 6.8. ● TiO$_2$, ▽ ZnO, ■ SiO$_2$.

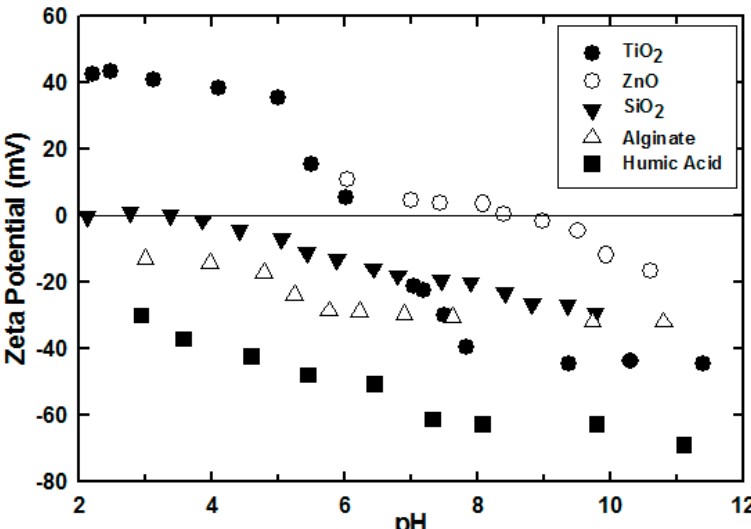

**Figure 4.** Zeta potential vs. pH of ENP particles (● $TiO_2$, ○ ZnO, ▼ $SiO_2$). Experimental conditions: $TiO_2$ = 50 mg/L, ZnO = 50 mg/L, $SiO_2$ = 1 mg/L, Alginate = 100 mg/L, Humic Acid = 100 mg/L.

According to the DLVO theory, the range of the surface charge spans up to 50 nm from the particle surface, which is the widest among the existing forces. The wide range will be the initial force attracting the ZnO to sludge particles and would be what results in higher attachment compared to $TiO_2$.

Silicon dioxide was selected due to its ubiquitous use in paints, cosmetics, and other household products. Results showed silicon dioxide with the highest amount of removal, showing two degrees' magnitude higher than that of ZnO, and three degrees of magnitude higher than $TiO_2$. Unlike the other particles, $SiO_2$ displays a negative charge, making it undesirable for attachment, according to the DLVO theory. However, the higher amount of silicon dioxide removal may be due to particle hydrophilicity, and its aggregation characteristics. Fumed silica is known to form large sizes of aggregates in aquatic media, forming a network of agglomerates. With the existence of materials other than silica, agglomerates of heterogeneous nature, including silica and other materials, will form. With the existence of large organic matter with fumed silica, it can be deduced that the removal of the silica particles is mainly due to sedimentation and agglomerates of silica and organic materials. It can be seen in Figure 3 that the removal of $SiO_2$ from the wastewater samples is more linear compared to the other ENPs, showing that $SiO_2$ does not follow a sorption isothermal removal. Additionally, due to the high salinity conditions of wastewater, the particles may also experience a salt-out effect, increasing sludge–particle interaction as well as particle–particle aggregation [47]. Image analysis of the three different ENPs (Figures S5–S7) also indicated similar results with the sorption results in Figure 3. $TiO_2$ was mainly observed as individual particles or in small groups, when attached to sludge particulates, ZnO was attached evenly to the surface of sludge flocs, and $SiO_2$ particles were observed in the sludge in large groups.

### 3.3. Effect of MLSS Concentration on Engineered Nanoparticle Sorption

To verify the effects of sludge concentration on ENP attachment, sorption experiments were conducted under various MLSS concentrations. Sorption experiments with $TiO_2$ and ZnO were tested on both primary and secondary sludge of various MLSS concentrations (Figures 5 and 6). Results in Figures 5 and 6 showed higher degrees of attachment occurring under lower MLSS concentrations. Although the higher ENP to organic ratio may contribute to the results, the fact that results showed up to four degrees of magnitude difference in attachment indicates various phenomena contributing to the interactions. ENP sorption to high MLSS concentrations is influenced by a number of factors, including DOM attachment to ENP, organic floc size, and sedimentation velocity.

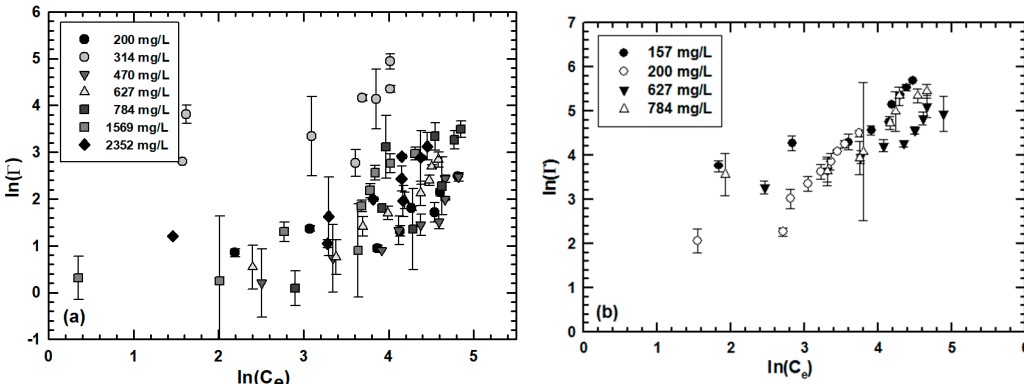

**Figure 5.** Attachment of TiO$_2$ to wastewater sludge of various concentrations. (**a**) Primary WSM; (**b**) Secondary WSM. Experimental conditions: Reaction time 5 h. pH = 6.8.

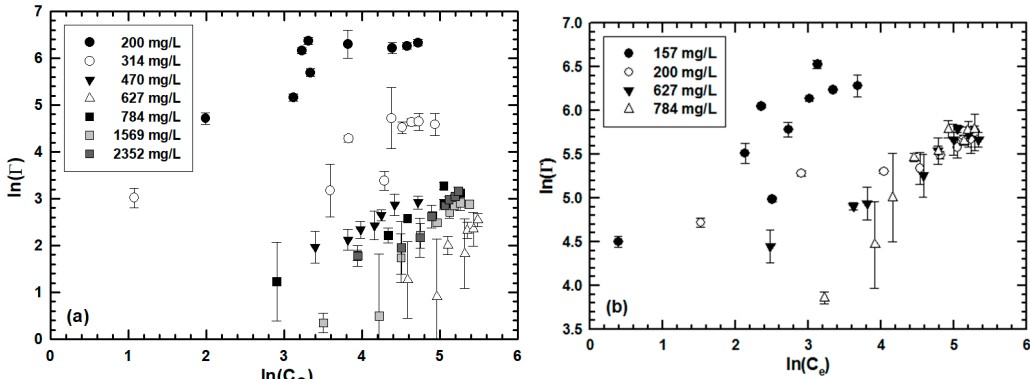

**Figure 6.** Sorption of ZnO to wastewater sludge of various concentrations. (**a**) Primary WSM; (**b**) Secondary WSM. Experimental conditions: Reaction time 5 h. pH = 6.8.

Under high organic loads, the sorption of DOM to ENP will stabilize the ENP. As has been proven in previous research, steric hindrance of organic matter stabilizes the ENP, increasing the critical coagulation concentration (CCC) value up to three degrees of magnitude. DOM such as natural organic matter, organic acids, and extracellular polymeric substances display negative functional groups that aid in steric repulsion [48,49], resulting in the hindrance of DOM-coated ENP and organic floc interaction. A high concentration of organic matter also results in higher viscosity, where the viscosity is known to increase exponentially with organic loading [50]. This may lead to the reduction of particle mobility of ENP, resulting in decreased attachment to sludge. With low organic loadings, higher ENP attachment occurred due to higher particle collision frequency and organic bridging [51]. Comparatively, higher ENP concentrations led to a higher frequency in particle–particle interaction and lower particle–DOM interaction. The interaction will result in larger particle aggregates that attach to organic flocs showing higher attachment results.

The floc size is also a factor in ENP sorption capacity, with limited organic concentrations, the increase in floc diameter will result in smaller surface areas. In addition, with larger organic flocs, the settling velocity of the flocs will increase, resulting in less time for ENP attachment. It has been proven that the addition of ENP would enhance the aggregation of organic flocs, where ENP attachment reduces the internal energy barrier and leads to an enhancement in the flocculation [52].

As can be seen in Figures 5b and 6b, both ZnO and TiO$_2$ show a stronger affinity to secondary sludge, which are two to three degrees of magnitude higher compared to primary sludge (Figures 5a and 6a). The higher affinity to secondary sludge is due to the organic composition, and viscosity of the wastewater samples. Primary sludge is composed of various organic acids, sugars, and fatty acids, whereas secondary sludge is composed of flocculated microbial aggregates [53,54]. Due to the larger amounts of dissolved organic

matter and lower amounts of flocculate matter, particles in the primary tank are more likely to interact with the dissolved matter, resulting in the stabilization of the ENP. Additionally, with the high organic and fatty acid concentration, primary wastewater samples display higher viscosity, resulting in slower ENP transport toward flocculent matter. On the other hand, lower viscosity and higher floc concentrations in the secondary wastewater will aid in the attachment of ENP onto and into the flocs, resulting in a higher degree of nanoparticle removal.

Aside from sludge concentration and characteristics, the charge of the ENP also contributes to the attachment characteristics. By comparing the results in Figures 5 and 6, it can be seen that Freundlich constants of ZnO (Figure 6) are more distinctively correlated with the sludge concentration than $TiO_2$ (Figure 5). As mentioned above, the ZnO particles display higher dispersion characters under the pH conditions of the wastewaters, resulting in more even distribution onto the sludge.

### 3.4. Effect of Ionic Strength on Engineered Nanoparticle Uptake

The influence of ionic strength (IS) on the interaction of ENP and organic matter was investigated with secondary WSM. Figure 7 shows the attachment of $TiO_2$ to WSM under NaCl concentrations ranging from 280 mM to 830 mM, with the BET sorption curves overlapped on the experimental data. With the increase of ionic strength (IS), the overall sorption showed a decrease in ENP attachment. The calculated BET parameters are shown in Table 1, indicating the stronger affinity of adsorption as well as higher maximum sorption capacities ($\Gamma_m$), with lower IS. This can be explained by the size and surface area of the organic flocs, where higher ionic strength reduces the repulsion of linear organic material resulting in its physical curling. The curling of the organic matter will form denser and more compact flocs, resulting in an overall lower amount of surface area. It has also been proven that with higher ionic strength, organic flocs have a tendency to grow in an elongated shape, making it more likely for flocs to collide and attach [47]. This is different from the organic curling effect mentioned above, where higher ionic strengths curl up the DOM and small organic matter, and the attachment of the flocs forms in an elongated shape. As a result of the larger flocs, the surface area per organic content decreases, lowering the number of surface sites for ENP attachment. Additionally, with higher salt concentrations the boundary water content of sludge also decreases, lowering the viscosity of the flocs [48,50]. This may additionally aid in the increase of settling velocities, shortening the time for ENP to attach to sludge surfaces.

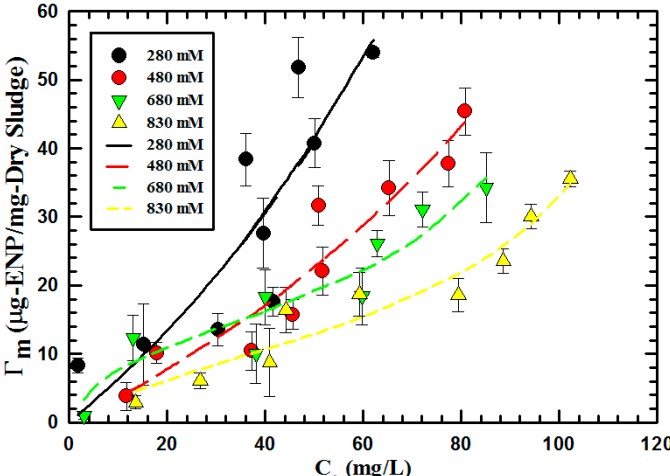

**Figure 7.** Effect of ionic strength on the adsorption of $TiO_2$ onto sludge particulates with BET analysis. Experimental conditions: pH = 6.84, MLSS = 2186 mg/L, electrolyte = NaCl, Secondary WSM.

**Table 1.** Summary of BET and Dual Langmuir sorption parameters of TiO$_2$ to WSM, under various ionic strength conditions.

| | NaCl (mM) | 280 | 480 | 680 | 830 |
|---|---|---|---|---|---|
| BET | $\Gamma_m$ ($10^{-10}$ mol-ENP/mg-Dry sludge) | 210.03 | 121.12 | 60.59 | 9.48 |
| | $K_1$ ($10^{10}$ M$^{-1}$) | 0.0028 | 0.0073 | 0.016 | 0.045 |
| | $K_2$ ($10^{10}$ M$^{-1}$) | 0.0051 | 0.0053 | 0.0074 | 0.0083 |
| | $R^2$ | 72.51 | 96.78 | 99.51 | 93.46 |

On the contrary, with lower IS, organic materials form linear shapes, having a tendency to extend and detach from the flocs due to higher electrostatic repulsion [55]. This will result in the breakage of flocs into smaller aggregates, encouraging TiO$_2$ sorption through floc entrapment during the rearrangement process. Additionally, with linear organic material, bridging of particular matter may also occur, increasing particle attachment. The combined effects show that low ionic strength may result in a multi-layer sorption model. Similar cases have been reported, where TiO$_2$ may display up to 5-layer sorption to algal cells [56].

In Table 1, the BET analysis also shows an increase of the $K_1$ constant with the increase of IS. The $K_1$ constant indicates the sorption of the first layer, showing a higher organic to ENP affinity with the increasing IS, which can be explained by the DLVO theory. With the increase of ionic strength, the electric double layer is compressed, where the hydrophobic attraction is unchanged. This leads to conditions where hydrophobic attraction becomes the dominating factor of TiO$_2$ and organic interactions [42,47]. With the hydrophobic interaction, TiO$_2$ displays a strong affinity to organic flocs as well as DOM [56]. The increased attachment will also influence the nanoparticles physically, where the attachment of DOM to TiO$_2$ surfaces will result in lower diffusion of the nanoparticles. The diffusion is a result of the increase in particle diameter, where the attached DOM retards the particle diffusion.

The $K_2$ values, which indicate the adsorption of additional layers, also increase with the IS, although it is at a lower degree than that of the $K_1$ values. This can also be explained by the DLVO theory, where the increase of $K_2$ values is due to the attachment of free nanoparticles with the initial TiO$_2$ layer attached to the organic flocs. The increase in IS compresses the electric double layer on the TiO$_2$ attached to the organic flocs, making the free TiO$_2$ particles more favorable for attachment. The increase of the $K_2$ value is lower than that of the $K_1$ value due to the low charge of the TiO$_2$ under neutral pH conditions of the wastewater samples.

The focus of the current research was to understand the removal of ENPs in wastewater treatment plants. Due to the fact that the majority of the ENPs are removed in the sludge processes, sorption characteristics of ENPs to primary and secondary sludge under various aquatic conditions were tested. To investigate the complex interactions and environmental factors that influence the sorption, different types of ENPs were tested under various MLSS concentrations and salinity conditions. Results showed that the sorption of ENP to sludge is a complex process of various interactions. Testing ENPs of different characteristics showed that the surface charge of ENP has a strong impact on the attachment to sludge, where the positively charged ZnO particles have a higher sorption capacity compared to neutral TiO$_2$ particles. Effects of hydrophobic/phallic interactions as well as the agglomeration of the particles will highly influence the removal, as seen with fumed SiO$_2$ particles. MLSS concentrations also showed an impact on the removal of ENPs, where results showed higher degrees of ENP attachment with lower MLSS concentrations due to larger surface areas of the smaller flocs. ENPs also showed a higher affinity towards secondary sludge due to the higher organic and microbial composition as well as higher floc concentration. With the increase of ionic strength (IS), results showed a decrease in the sludge's maximum sorption capacity on ENP. This is a combined effect of the physical curling of the flocs and the compression of the double layer of the ENPs under high salinity conditions.

The current research presents the sorption of ENP to primary and secondary sludge under various conditions to verify the effects and mechanism of the sorption process.

However, due to the preliminary stage of the research, ENP interactions were researched with mainly the bulk of the sludge. Thus, the individual factors such as the composition of organic material of sludge and ENP surface interactions are still unknown. Additional research should be conducted on the interaction of ENPs and various organic compounds to better understand the phenomena.

**Supplementary Materials:** The following supporting information can be downloaded at: https://www.mdpi.com/article/10.3390/w15162872/s1, Figure S1: Sedimentation profile of Zn and $TiO_2$ with DOM. Primary DOM = 400 mg/L, Secondary DOM = 250 mg/L. Reaction time 5 h. pH = 6.8. The x–axis indicates the sedimentation time, where the y-axes (TS) indicates the turbidity of the sludge (a) Secondary DOM + $TiO_2$. (b) Primary DOM + ZnO (c) Secondary DOM + ZnO; Figure S2: Sedimentation profile of ZnO and $TiO_2$ with supernatant (small organic matter < 1 μm). Primary supernatant = 567 mg/L. Reaction time 5 h. pH = 6.8 (a) Secondary supernatant + $TiO_2$. (b) Primary supernatant + ZnO (c) Secondary supernatant + ZnO; Figure S3: Sedimentation profile of Zn and $TiO_2$ with large organic particles (>1 μm). Primary = 2242 mg/L, Secondary = 6100 mg/L. Reaction time 5 h. pH = 6.8 (a) Primary + ZnO (b) Secondary + $TiO_2$. (c) Secondary + ZnO; Figure S4: Sedimentation profile of Zn and $TiO_2$ with sludge. Primary = 2942 mg/L, Secondary = 6667 mg/L. Reaction time 5 h. pH = 6.8 (a) Primary sludge + ZnO (b) Secondary sludge + $TiO_2$. (c) Secondary sludge + ZnO; Figure S5. SEM image and EDS analysis of $TiO_2$ sorption to sludge particulates. (a) SEM image of $TiO_2$ attached to sludge particulates, (b) elemental analysis of Titanium from the SEM image from image (a), (c) Energy dispersive spectroscopy (EDS) analysis of the particular matter in image (a); Figure S6. SEM image and EDS analysis of ZnO sorption to sludge particulates (a) SEM image of ZnO attached to sludge particulates, (b) elemental analysis of Zinc from the SEM image from image (a), (c) Energy dispersive spectroscopy (EDS) analysis of the particular matter in image (a); Figure S7. TEM image and EDS analysis of $SiO_2$ sorption to sludge particulates (a) TEM image of $SiO_2$ attached to sludge particulates, (b) elemental analysis of silica from the TEM image from image (a), the green images indicate organic matter where orange images indicate elemental silica.

**Author Contributions:** Conceptualization, C.-P.H. and M.J.; Writing—Original draft preparation, S.C.; Investigation, C.-P.H. and S.C.; Supervision, C.-P.H.; Validation, G.-S.W. All authors have read and agreed to the published version of the manuscript.

**Funding:** This work was supported by U.S. Environmental Protection Agency through a grant no. R8348590. Additional support was provided by Chungnam National University, Daejeon, South Korea grant no 2021-0242-01.

**Data Availability Statement:** Not applicable.

**Conflicts of Interest:** The authors declare no conflict of interest.

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
