# Peer review of "The Uptake of Engineered Nanoparticles by Sludge Particulates"

_water, doi:10.3390/w15162872_

Round 1
Reviewer 1 Report
The authors investigated the adsorption of typical engineered nanoparticles (i.e., SiO2, ZnO, and TiO2) on to the fractionated sludges from WWTP using the differential sedimentation method. The impacts of MLSS, ionic strength, and nanoparticle concentration were further studied. Although the present results are interesting and insightful, the current version of manuscript is not well organized and methodology and conclusions parts need a large improvement.
1) Overall structure: “Results” section has almost no content, and it is unclear which specific result the present description is related to. I highly recommend to integrate “Results” and “Discussions” as “Results and Discussions”.
2) Methodology: “Materials and Methods” section is not well organized and informative. Please describe the condition of sonication (L96), sampling dates (L100), method of MLSS measurement (L105), method of elemental analysis (L106), the concentrations of NaCl, HCl, NaOH solutions (L136-137), method of TOC measurement (L278-279). Also, the sorption models examined in the discussions part (ex. BET and Dual Langmuir) were not explained in the method.
3) Methodology: In my understanding, “Differential sedimentation (L138-178)” section describes the rationale for why the authors used the differential sedimentation method, and the detailed procedure of the method is described in the following “Analysis of sedimentation curves” section. Am I correct? If yes, this rationale is too long and redundant. I recommend to integrate “Differential sedimentation” and “Analysis of sedimentation curves” with rewriting the rationale part more concisely.
4) Methodology: The title of the section, “Sedimentation experiment”, is not appropriate since other sections are also related to sedimentation experiment. Please revise it to the more specific title.
5) Conclusions: The current version of manuscript doesn’t mention the key findings/highlights of the study nor the limitation of the study. Furthermore, please suggest the more detailed direction of the future research (to overcome the limitation of the present study), instead of just saying “additional research will be needed”.
Minor points
1) “ ..?? m from the sludge.(L127)” ?? is the garbled text.
2) Please spell out the abbreviated terms such as BOD or TOC, when they are first used.
Author Response
Point 1: Overall structure: “Results” section has almost no content, and it is unclear which specific result the present description is related to. I highly recommend to integrate “Results” and “Discussions” as “Results and Discussions”.
Response 1: Page 8 Line 306, We strongly agree with the reviewers with the fact on the lack of contents in the “Results” section. The two sections were combined as “Results and Discussions”. Thank you.
Point 2: Methodology: “Materials and Methods” section is not well organized and informative. Please describe the condition of sonication (L96), sampling dates (L100), method of MLSS measurement (L105), method of elemental analysis (L106), the concentrations of NaCl, HCl, NaOH solutions (L136-137), method of TOC measurement (L278-279). Also, the sorption models examined in the discussions part (ex. BET and Dual Langmuir) were not explained in the method.
Response 2: We agree with the reviewer on the fact that some of the information regarding the “Material and Methods” lack critical points. Regarding the points that the reviewer has pointed out, the following corrections have been made.
Page 3 line 101~103
Information on the operating conditions of the high-powered sonicator has been provided as “high-powered sonicator at 100W for 10 sec (Ultrasonic Homogenizer 4710 Series, Cole-Parmer Instrument Co., Chicago, IL)”
Page 3 Line 110~112
The method of MLSS measurement has been stated in the paper. The method used is based on the method stated in the “Standard Methods for the Examination of Water and Wastewater”, prococol 2540D.
Page 3 Line 116
The analysis method for the elemental analysis has been stated in the paper, where explanation of the analysis machine was added.
Page 4, Line 144-1145
The stock solution of the NaCl was stated in the paper, and for the HCl and NaOH, staock solutions were not made for the chemicals instead diluted and added to the samples. An explanation on the chemical grade and supplier has been added.
Page 7 Line 260~263
The Method, analysis machine was added to the context regarding the TOC analyzer.
Page 8, Line 297-303
Additional explanations of the BET analysis has been added to the paper.
The dual Langmuir analysis has been considered to not provide novel findings compared to the BET analysis, resulting in the removal of the section regarding the dual Langmuir analysis. For so, explanations (equations) on the method has not been added.
Point 3: Methodology: In my understanding, “Differential sedimentation (L138-178)” section describes the rationale for why the authors used the differential sedimentation method, and the detailed procedure of the method is described in the following “Analysis of sedimentation curves” section. Am I correct? If yes, this rationale is too long and redundant. I recommend to integrate “Differential sedimentation” and “Analysis of sedimentation curves” with rewriting the rationale part more concisely.
Response 3: We agree with the reviewer on the long explanation of context in the “Differential sedimentation” section. The “Differential sedimentation” and “Analysis of sedimentation curves” were combined and rewritten more concisely for readers.
Point 4: Methodology: The title of the section, “Sedimentation experiment”, is not appropriate since other sections are also related to sedimentation experiment. Please revise it to the more specific title.
Response 4: The reviewer has a valid point on the title of the “Sedimentation experiment” is confusing with other sections in the methodology. The title has been changed to “ENP sorption experiment and evaluation” for a more specific title to explain the section. (Page 7, Line 264)
Point 5: Conclusions: The current version of manuscript doesn’t mention the key findings/highlights of the study nor the limitation of the study. Furthermore, please suggest the more detailed direction of the future research (to overcome the limitation of the present study), instead of just saying “additional research will be needed”.
Response 5: We agree with the reviewer on the lack of key findings/highlights of the study. To tackle this issue, the key findings/highlights were summarized in the final section of the “Results and Discussion”. (Page15 Line553~Page16 Line 578). The major findings were summarized as well as the imitations of the study, and ending with future research suggestions.
Point 6: “ ..?? m from the sludge.(L127)” ?? is the garbled text.
Response 6: We thank the reviewer for finding the error in the paper. We have fixed the units to ‘μm’. (Page 3, Line 128)
Point 7: Please spell out the abbreviated terms such as BOD or TOC, when they are first used.
Response 7: We thank the reviewer for pointing out the lack of explanations of the abbreviated terms. The abbreviated terms are spelled out as followed.
Page 2, Line 54
BOD was fixed to biochemical oxygen demand (BOD).
Page 7 Line 261
TOC was fixed to total organic carbon (TOC).
Please see the attachment

Reviewer 2 Report
After reading the manuscript (ms) presented by Choi et al., I have some major comments that the authors must clarify:
- What is the polymorph phase for the TiO2 NPs used in this work? Is it referred to as pure anatase, brookite, or a mixture? Please, this must be added somewhere in the ms.
- Please add the next reference [Ecotoxicological Properties of Titanium Dioxide Nanomorphologies in Daphnia magna] that agrees with the point of zero charge of the here studied TiO2 NPs, located at 6.5! This can be helpful to respond to query 1.
- Despite the author's purchase of the NPs? Some comments about confirmation of the involved phases and crystallinity are not given in the ms? It means that without a rigorous characterization, how do the authors know that pure phases have been purchased? How can this affect their determination in the sludge?
Minor points:
In line 128, what is the meaning of the unidentified symbol?
Some equations are in bold; it is necessary.
The introduction must cite recent articles. Please update it.
Author Response
Point 1: What is the polymorph phase for the TiO2 NPs used in this work? Is it referred to as pure anatase, brookite, or a mixture? Please, this must be added somewhere in the ms.
Response 1: We agree that the polymorphology of the TiO2 particles should be added to the paper. Hence, in the material and method section, an explanation of the configuration of the P25 was given with additional references to prove the crystalline structure of the TiO2 used in the study.
The references given in the paper are the paper bellow. (Both papers analyze the P25 with XRD analysis in the paper)
Xiongzhen Jiang, Maykel Manawan, Ting Feng, Ruifeng Qian, Ting Zhao, Guanda Zhou, Fantai Kong, Qing Wang, Songyuan Dai, Jia Hong Pan. Anatase and rutile in evonik aeroxide P25: Heterojunctioned or individual nanoparticles? Catalysis Today 2018, 300, 12-17
Marcela Chaki Borrás, Ronald Sluyter, Philip J. Barker, Konstantin Konstantinov, Shahnaz Bakand. Y2O3 decorated TiO2 nanoparticles: Enhanced UV attenuation and suppressed photocatalytic activity with promise for cosmetic and sunscreen applications. Journal of Photochemistry and Photobiology B: Biology 2020, 207, 111883.
Point 2: Please add the next reference [Ecotoxicological Properties of Titanium Dioxide Nanomorphologies in Daphnia magna] that agrees with the point of zero charge of the here studied TiO2 NPs, located at 6.5! This can be helpful to respond to query 1. .
Response 2: We thank the reviewers for recommending a paper that will strengthen our logic on explaining the zero point charge of the TiO2 and its correlation with its crystalline structure. The reference given by the reviewers has been added to the paper in the “Effect of Engineered Nanoparticle Type” explaining the zero point charge and its influence on the interaction with wastewater sludge (Line 411~414)..
Point 3: Despite the author's purchase of the NPs? Some comments about confirmation of the involved phases and crystallinity are not given in the ms? It means that without a rigorous characterization, how do the authors know that pure phases have been purchased? How can this affect their determination in the sludge?
Response 3: We understand the concern of the reviewer on the composition of the TiO2 used in the study which should rightfully be questioned. However, due to the focus of the study which is the interaction of nanoparticles in wastewater treatment plants, we think that purchased TiO2 with a variety of different compositions would be more fitting for the study. Since waste water contains a wide variety of TiO2 used in all types of consumer products, so to properly represent nanoparticles that are found in field conditions of wastewater treatment plants were needed and P25 was (in our opinion) the best fit for our research.
Point 4: In line 128, what is the meaning of the unidentified symbol?
Response 4: We thank the reviewer for finding the error in the paper. We have fixed the units to ‘μm’. (Page 3, Line 128)
Point 5: Some equations are in bold; it is necessary
Response 5: We thank the reviewer for pointing out the inconsistency in the details in the equations. All the equations in the paper was changed from bold to regular thickness.
Point 6: The introduction must cite recent articles. Please update it.
Response 6: We agree with the reviewer that most of the papers that are referenced in the research are out dated. A number of papers regarding current research on the topic has been added to the paper spanning from 2016 to 2022.
Please see the attachment

Reviewer 3 Report
This article focuses on how a variety of commonly engineered nanoparticles (ENPs) interact as a function of time with sludge/water taken from an active wastewater treatment plant (WWTP). The authors use the WWTP sludge/water under different environmental conditions such as concentration, pH, salt concentration, and size of sludge particles to replicate how the ENPs will attach to the sludge particles during the WWTP process. This article highlights an important topic that will become ever more increasingly important as we as a global society incorporate synthesized nanoparticle into many over the counter everyday products. Below are a few recommendations for the authors:
1. Line 17, in the abstract, the word ‘simulate’ is used. This may be replaced with ‘replicate’ if desired to eliminate any confusion as a pure simulation work by the novice reader.
2. The abstract seems to inform the reader that SiO2, ZnO, and TiO2 will be examined equally. This appears not to be the case as SiO2 is not mentioned until well past the halfway point in the manuscript (page 11 of 19 line 408 of 697). Maybe a separate sentence is needed in the abstract to describe the different experiments with SiO2; thus highlight the SiO2 displays a negative charge, unlike the other particles, and does not favor attachment. This is not mentioned until lines 433-434. This information can easily be included briefly in the abstract. Explain in the abstract SiO2 results will come later in the article.
3. ZnO, and TiO2 are thoroughly discussed in the Introduction. SiO2 has been completely left out of the Introduction. Some information about SiO2 nanoparticles in WWT should be mentioned, otherwise it looks like SiO2 was simply forgotten in the Introduction and the paper until the reader passes the halfway mark. In the Introduction, the negative charge with SiO2 could be discussed and why the turbidity tests where not done in the same fashion as ZnO, and TiO as a result.
4. At the beginning of the Results section, combined with the information presented in the Introduction, the authors highlight that the literature tells them the ENPs are mostly found in the primary and secondary sludge, but do mention 26-39% possibly make it back into the environment. Understanding how to reduce this number that make it back to the environment through the attachment of the ENPs to sludge is important. However, the uptake of high concentrations of ENPs in the sludge might also become problematic for WWTPs facilities if regulations are put in place that put restrictions on the amount of ENPS that can be in sludge. Can references be included that discuss the cost of remediation of ENPs from the sludge for WWTPs before be sold or landfilled? Are there references to legislation being put into place or that currently exists with the aim of monitoring the levels of ENPs in the solids/sludge before becoming such items as fertilizers (or landfilled)? Such references could draw more attention to this work and highlight the critical issues.
5. Line 95, the word ‘prior’ does not need capitalized.
6. Line 110, the word ‘negligible’ could be accompanied by a number to provide meaning to the word ‘negligible’.
7. Line 127-128, there seems to be an error in the format for the units.
8. On line 152 and 176, the authors refer to the phrase ‘current chapter’, which seems out of place for an article in a journal.
9. In lines 165-167, the authors may want to refer to Figure 1 as an example to go along with the description presented there.
10. In the beginning of the Analysis section, starting line 180, the authors may want to provide the novice reader more of a description of ‘differential sedimentation’ in addition to providing the reference they have.
11. Line 168, there appears to be a misplaced comma.
12. Line 194-195 the authors assume spherical particles. Are there references that indicate a valid assumption? If not what is the expected error that may result from this assumption.
13. Figure 1 seems to come too early in the manuscript. Questions arise when seeing Figure 1 first, before the text describing Figure 1, which appears to come later. Figure 1 could easily be placed after the paragraph that contains Equation (7), thus before line 238.
14. Figure S1 seems to be identical to the data used in Figure 1. Figure 1 provides more information than Figure S1 via additional fits. Why is Figure S1 needed? Figure S1 could be removed and the Supporting Figures could be renamed and correctly mentioned throughout the manuscript. Thus the reference to S1 on line 249 would be to Figure 1, with all following numbering changed throughout the paper.
15. Line 260, there appears to be a missing period before ‘Figure 1’.
16. The discussion of how the data was collected through UV-visible spectroscopy and then converted to turbidly (lines 287-289) could be discussed early on, before the reader see Figure 1 which has turbidity on the y-axis.
17. Figure 2. I see the authors are taking to maximize the data shown in the given space. Good. However, Figure 2 (b) could be set to 2.0 on the y-axis like Figure 2 (c) and 2 (d). Likewise the maximum on Figure 2 (a) could be set to the nearest integer like the other figure panels.
18. The ‘BET’ acronym first used in Figure caption 7 Line 505 is never defined in the manuscript and needs to be. This ‘BET’ is used many other places.
19. Line 530, the word ‘matter’ seems to missing a ‘t’.
20. Line 536, the ‘K1’ and ‘K2’ could be briefly defined and described before diverting the reader to DLVO theory.
21. The authors should confirm why the labels on the y-axis are different in the supplementary figures. S1 has the y-axis as Turbidity and is referred to as a sedimentation curve in the manuscript. S2 has TS and is referred to as a sedimentation profile. State what are the differences in the y-axis to require a different label. Figure S2 in the manuscript uses Turbidity, but figures similar in the Supplemental section use TS for the y-axis. Provide an explanation to the reader.
22. The legend for Figure 2 (b) is smaller than for the other panels in the figure.
23. Figure S2 is missing panel labels (a), (b), and (c).
24. Figure S3 is missing panel labels (a), (b), and (c).
25. Figure S4 is missing panel labels (a), (b), and (c).
26. Figure S5 is missing panel labels (a), (b), and (c).

Author Response
Point 1: Line 17, in the abstract, the word ‘simulate’ is used. This may be replaced with ‘replicate’ if desired to eliminate any confusion as a pure simulation work by the novice reader
Response 1: We agree with the reviewer on the usage of the word “replicate” in the place of “simulate’. The context of the abstract has been fixed accordingly
Point 2: The abstract seems to inform the reader that SiO2, ZnO, and TiO2 will be examined equally. This appears not to be the case as SiO2 is not mentioned until well past the halfway point in the manuscript (page 11 of 19 line 408 of 697). Maybe a separate sentence is needed in the abstract to describe the different experiments with SiO2; thus highlight the SiO2 displays a negative charge, unlike the other particles, and does not favor attachment. This is not mentioned until lines 433-434. This information can easily be included briefly in the abstract. Explain in the abstract SiO2 results will come later in the article.
Response 2: We agree with the reviewer that the explanation in the abstract can be misleading where it may seem that the sedimentation experiments were conducted on all three ENPs (TiO2, ZnO, SiO2). The abstract was rewritten, to inform that the sedimentation experiments were conducted with TiO2 and ZnO. Experiments regarding SiO2 were separately mentioned later in the abstract highlighting the results.
Point 3: ZnO, and TiO2 are thoroughly discussed in the Introduction. SiO2 has been completely left out of the Introduction. Some information about SiO2 nanoparticles in WWT should be mentioned, otherwise it looks like SiO2 was simply forgotten in the Introduction and the paper until the reader passes the halfway mark. In the Introduction, the negative charge with SiO2 could be discussed and why the turbidity tests where not done in the same fashion as ZnO, and TiO as a result.
Response 3: Thank you for pointing out the fact that information on SiO2 was excluded from the introduction. Information on SiO2 and its impact on wastewater treatment processes as well as its impact on soil microbe diversity has been added to the introduction. Additionally, the rational of why the sedimentation experiments were conducted mainly on TiO2 and ZnO were also stated in the introduction.
Point 4: At the beginning of the Results section, combined with the information presented in the Introduction, the authors highlight that the literature tells them the ENPs are mostly found in the primary and secondary sludge, but do mention 26-39% possibly make it back into the environment. Understanding how to reduce this number that make it back to the environment through the attachment of the ENPs to sludge is important. However, the uptake of high concentrations of ENPs in the sludge might also become problematic for WWTPs facilities if regulations are put in place that put restrictions on the amount of ENPS that can be in sludge. Can references be included that discuss the cost of remediation of ENPs from the sludge for WWTPs before be sold or landfilled? Are there references to legislation being put into place or that currently exists with the aim of monitoring the levels of ENPs in the solids/sludge before becoming such items as fertilizers (or landfilled)? Such references could draw more attention to this work and highlight the critical issues.
Response 4: I am sorry to report that a scares amount of research is conducted on the remediation of ENP from sludge. There are studies on remediation methods that utilize nanoparticles, however, the researched methods do not state on how the nanoparticels used for remediation is to be retrieved, which just intensifying the issue. We have also been looking into the regulations regarding the prevention of nanopartciels entering the landfills for lack of success. We thank the reviewer on enlightening us on the issue and will pursue more information and activity toward the legislation on the given matter. Thank you.
Point 5: Line 95, the word ‘prior’ does not need capitalized.
Response 5: We thank the reviewer for pointing out the mistake in the paper. The word ‘prior’ has been fixed to a lower case. (Page 3, Line 95)
Point 6: Line 110, the word ‘negligible’ could be accompanied by a number to provide meaning to the word ‘negligible’.
Response 6: We agree with the reviewer that it is important to state the concentration of the detected elements, and have included the detected concentration of both Ti and Zn from the sludge that was used in the experiments. (Page 3, Line 110)
Point 7: Line 127-128, there seems to be an error in the format for the units.
Response 7: We thank the reviewer for finding the error in the paper. We have fixed the units to ‘μm’. (Page 3, Line 128)
Point 8: On line 152 and 176, the authors refer to the phrase ‘current chapter’, which seems out of place for an article in a journal.
Response 8: We agree with the reviewer that the phrase ‘current chapter’ is off-putting. We have removed the phrases and rewritten the chapter. (Page 4, line 157~169)
Point 9: In lines 165-167, the authors may want to refer to Figure 1 as an example to go along with the description presented there.
Response 9: We agree with the reviewer on the fact that the explanation of the differential sedimentation curve would be better understood by referring to Figure 1. Reference to Figure 1 was added to the phrase in Page 4 line 167.
Point 10: In the beginning of the Analysis section, starting line 180, the authors may want to provide the novice reader more of a description of ‘differential sedimentation’ in addition to providing the reference they have.
Response 10: We agree with the reviewer on the tedious explanation process that may prevent the novice readers from understanding the main focus of the chapter which is the explanation of the differential sedimentation. For a better explanation of the approach, the tedious explanation of the background was removed and rewritten with a focus mainly on the approach of the method.
Point 11: Line 168, there appears to be a misplaced comma.
Response 11: We thank the reviewer for finding the misplaced comma in the paper. The comma has been replace for more accurate grammar. Thank you. (Page 4, Line 168)
Point 12: Line 194-195 the authors assume spherical particles. Are there references that indicate a valid assumption? If not what is the expected error that may result from this assumption.
Response 12: We agree that stating the shape of the ENPs with only an assumption. For a better explanation for the shape of the ENPs, SEM images of the particles attached to the sludge was added to the paper via supporting information. (Figure S5-S7)
Point 13: Figure 1 seems to come too early in the manuscript. Questions arise when seeing Figure 1 first, before the text describing Figure 1, which appears to come later. Figure 1 could easily be placed after the paragraph that contains Equation (7), thus before line 238.
Response 13: We agree with the reviewer that Figure 1 is located too early in the paper. The figure has been relocated after Equation (7).
Point 14: Figure S1 seems to be identical to the data used in Figure 1. Figure 1 provides more information than Figure S1 via additional fits. Why is Figure S1 needed? Figure S1 could be removed and the Supporting Figures could be renamed and correctly mentioned throughout the manuscript. Thus the reference to S1 on line 249 would be to Figure 1, with all following numbering changed throughout the paper.
Response 14: We agree with the reviewer that Figure S1 is not necessary in the text when explaining the sedimentation curve. As mentioned by the reviewer, the texts that indicate ‘Figure S1 ‘ were all fixed to stage Figure 1.
Point 15: Line 260, there appears to be a missing period before ‘Figure 1’.
Response 15: We thank the reviewer for pointing out the error in the paper. The missing period has been added. Thank you. (Page 7, Line 260)
Point 16: The discussion of how the data was collected through UV-visible spectroscopy and then converted to turbidly (lines 287-289) could be discussed early on, before the reader see Figure 1 which has turbidity on the y-axis.
Response 16: We agree with the reviewer on the process of the turbidity measuring, should be located early on in the paper. The explanation of the turbidity measurement was mentioned in the ‘Differential sedimentation’ section (Page4 Line163~164) before explaining the equation.
Point 17: . Figure 2. I see the authors are taking to maximize the data shown in the given space. Good. However, Figure 2 (b) could be set to 2.0 on the y-axis like Figure 2 (c) and 2 (d). Likewise the maximum on Figure 2 (a) could be set to the nearest integer like the other figure panels.
Response 17: We agree with the reviewers that the scale of the y-axes in Figure 2 should be matched to easily compare the data under the different conditions. The y-axes in Figure 2(b) was scaled from -1.0 to 2.0, and Figure 2(b) was also scaled from -10 to 2.0 to match the scale of the other figures.
Point 18: The ‘BET’ acronym first used in Figure caption 7 Line 505 is never defined in the manuscript and needs to be. This ‘BET’ is used many other places.
Response 18: We agree with the reviewer on the lack of clarification on the nomenclature of the acronym ‘BET’. The proper correction has been mate as so.
Page 8, Line 297
An explanation of the BET sorption equation has been added to the paper. And in the explanation process the term Brunauer-Emmett-Teller (BET) has been added.
Point 19: Line 530, the word ‘matter’ seems to missing a ‘t’.
Response 19: We thank the reviewer for pointing out the error in the paper. The missing ‘t’ has been added to the word ‘matter’. Thank you. (Page15, Line 530)
Point 20: Line 536, the ‘K1’ and ‘K2’ could be briefly defined and described before diverting the reader to DLVO theory.
Response 20: We agree with the reviewer that explanation of the term ‘K1’ and ‘K2” should be made to better explain the sorption phenomena. The corrections were conducted as so.
Page 15 line 535 and Line 545, Page 8 line 297-303
Explanations of the term ‘K1’ and ‘K2’ were conducted in line 522 as ‘The K1 constant indicate the sorption of the first layer’ and in line 532 as ‘The K2 values, which indicate the adsorption of additional layers’. A separate section was devoted to explain the BET sorption equation where the variables in the equations were each explained.
Point 21: The authors should confirm why the labels on the y-axis are different in the supplementary figures. S1 has the y-axis as Turbidity and is referred to as a sedimentation curve in the manuscript. S2 has TS and is referred to as a sedimentation profile. State what are the differences in the y-axis to require a different label. Figure S2 in the manuscript uses Turbidity, but figures similar in the Supplemental section use TS for the y-axis. Provide an explanation to the reader.
Response 21: We agree with the reviewer that explanations of the y-axes in Figure 2S should be stated in the paper. Explanations were added in the explanation in the figures. Additionally Figure S1 in the original version was removed due to the fact that it is better to explain the sedimentation process through Figure 1 than using Figure S1 (as stated in Point 14 in the review).
Point 22: The legend for Figure 2 (b) is smaller than for the other panels in the figure
Response 22: The legend in Figure 2 (b) has been fixed to match the size of the other legends in Figure 2 (a, c, d).
Point 23: Figure S2 is missing panel labels (a), (b), and (c).
Response 23: We thank the reviewer for pointing out the missing labels in Figure S2 (Currently Figure S1). We have added the labels accordingly.
Point 24: Figure S3 is missing panel labels (a), (b), and (c).
Response 24: We thank the reviewer for pointing out the missing labels in Figure S3 (Currently Figure S2). We have added the labels accordingly.
Point 25: Figure S4 is missing panel labels (a), (b), and (c).
Response 25: We thank the reviewer for pointing out the missing labels in Figure S4 (Currently Figure S3). We have added the labels accordingly.
Point 26: Figure S5 is missing panel labels (a), (b), and (c).
Response 26: We thank the reviewer for pointing out the missing labels in Figure S5 (Currently Figure S4). We have added the labels accordingly.
Please see the attachment

Reviewer 4 Report
Points to address for the authors:
Abstract: The lines 11-16 in the abstract should be reframed. I is vague and lacks coherence. The aim of the study should be clearly mentioned in the abstract.
The SiO2, TiO2, and ZnO nanoparticles should be characterized through SEM/TEM and their morphology needs to be discussed in detail. A DLS study will throw further light on the absorption capacity.
The study lacks this focus and can only be considered for publication if these points are addressed.
Author Response
Point 1: Abstract: The lines 11-16 in the abstract should be reframed. I is vague and lacks coherence. The aim of the study should be clearly mentioned in the abstract
Response 1: We agree with the reviewer on the lack of coherence in the abstract. Line 10~17 from the original abstract has been reframed to better explain the context of our research. (Page 1, Line 10~19).
Point 2: The SiO2, TiO2, and ZnO nanoparticles should be characterized through SEM/TEM and their morphology needs to be discussed in detail. A DLS study will throw further light on the absorption capacity.
Response 2: We agree with the reviewer that an image analysis including SEM or TEM images are to be added to the paper to not only characterize the shape of the ENPs but also its interaction with sludge. SEM/TEM images of the ENP attached to sludge has been added to the paper via supporting information with an explanation of the images in the main content of the paper in Line 435-439. Additionally, references on the crystalline structure of the TiO2 used in the research has also been added. In the research, TiO2 with anatase, rutile, and a small amount of amorphous structures was purchased and used due to the fact that all types of nanoparticles flow into wastewater treatment plants. (Since commercial products containing TiO2 are mainly composed of one of the three types of particles used in the research; line 95-96, 411-414).
Please see the attachment

Round 2
Reviewer 2 Report
The authors have addressed all my comments, and the manuscript can now be accepted for publication.
Reviewer 4 Report
The manuscript has been considerably improved. Though the subject matter is not novel, it might be considered for publication upon the editors approval.